# Radiomics in Triple Negative Breast Cancer: New Horizons in an Aggressive Subtype of the Disease

**DOI:** 10.3390/jcm11030616

**Published:** 2022-01-26

**Authors:** Camil Ciprian Mireștean, Constantin Volovăț, Roxana Irina Iancu, Dragoș Petru Teodor Iancu

**Affiliations:** 1Department of Oncology and Radiotherapy, Faculty of Medicine, University of Medicine and Pharmacy of Craiova, 200349 Craiova, Romania; mc3313@yahoo.com; 2Department of Surgery, Railways Clinical Hospital, 700506 Iasi, Romania; 3Department of Medical Oncology-Radiotherapy, Faculty of Medicine, “Gr. T. Popa” University of Medicine and Pharmacy, 700115 Iasi, Romania; volovatconstantin@gmail.com (C.V.); dt_iancu@yahoo.com (D.P.T.I.); 4Euroclinic Oncological Hospital, 700110 Iasi, Romania; 5Department of Oral Pathology, Faculty of Dentistry, “Gr. T. Popa” University of Medicine and Pharmacy, 700115 Iasi, Romania; 6Clinical Laboratory Department, “St. Spiridon” Emergency Hospital, 700111 Iasi, Romania; 7Department of Radiotherapy, Regional Institute of Oncology, 700483 Iasi, Romania

**Keywords:** radiomics, breast cancer, TNBC, biomarker

## Abstract

In the last decade, the analysis of the medical images has evolved significantly, applications and tools capable to extract quantitative characteristics of the images beyond the discrimination capacity of the investigator’s eye being developed. The applications of this new research field, called radiomics, presented an exponential growth with direct implications in the diagnosis and prediction of response to therapy. Triple negative breast cancer (TNBC) is an aggressive breast cancer subtype with a severe prognosis, despite the aggressive multimodal treatments applied according to the guidelines. Radiomics has already proven the ability to differentiate TNBC from fibroadenoma. Radiomics features extracted from digital mammography may also distinguish between TNBC and non-TNBC. Recent research has identified three distinct subtypes of TNBC using IRM breast images voxel-level radiomics features (size/shape related features, texture features, sharpness). The correlation of these TNBC subtypes with the clinical response to neoadjuvant therapy may lead to the identification of biomarkers in order to guide the clinical decision. Furthermore, the variation of some radiomics features in the neoadjuvant settings provides a tool for the rapid evaluation of treatment efficacy. The association of radiomics features with already identified biomarkers can generate complex predictive and prognostic models. Standardization of image acquisition and also of radiomics feature extraction is required to validate this method in clinical practice.

## 1. Introduction

Medical imaging is viewed as one of the top developments that have changed cancer care, as it has fundamentally changed the manner in which oncology physicians measure, manage, diagnose and treat cancer disease. Medical images have progressed substantially in recent decades, being a key player with an increasing role in the diagnosis and initial cancer evaluation, but also in the follow-up of the oncological disease. As the resolution of the images increased, a new science called radiomics was developed based on the computerized analysis of the data extracted from different types of medical images. Transforming high volumes of medical images into data that can be correlated with the prognosis and also can be used to predict the tumor response to treatment opens the horizons of using radiomics in the routine clinical setting. The concept is relatively new and the number of scientific papers describing applications of radiomics in medical practice has increased steadily since 2012, the year in which this concept was first proposed. The most commonly used imaging techniques for extracting features, in order to correlate them with clinical data, are computed tomography (CT), ultrasonography (US), magnetic resonance imaging (MRI) and positron emission tomography (PET-CT). Recently, the development of the molecular and genetic analysis of tumors has made possible the development of a new research direction based on correlating the data obtained from medical images with gene mutations, thus opening the way for a non-invasive evaluation of tumor genomic features. This new domain, characterized by a major development in recent years, based on the medical images linking to the tumor genomic and molecular profile, has been named “radiogenomics”. Being the second leading cause of cancer mortality worldwide, although advances in molecular biology have stratified breast cancer into prognostic molecular classes, the triple negative subtype is notable for its aggressiveness with a relatively limited therapeutic spectrum. An overall 4-year survival of approximately 77% for TNBC is lower than the OS of other breast cancer subtypes ranging from 82.7 to 92.5%. Radiomics is an essential player in the race to improve the management of TNBC from the identification of new biomarkers to therapeutic strategies, in the general context of the precision medicine concept implementation, in order to improve the prognosis of this relatively orphan disease in terms of less therapeutic progress in recent decades [1,2,3,4].

In contrast to the American College of Radiology Breast Imaging Reporting and Data System (BI-RADS) lexicon, first developed in 1993 to standardize communication between specialists involved in the diagnosis and treatment of breast cancer, radiomics offers us as quantification tools a series of features, some more intuitive (size and shape, spherical, asymmetry) others more abstract and harder to interpret for the clinician (kurtosis, entropy, skewness). Currently, the 5th edition of the BI-RADS^®^ atlas is a valuable tool for quantifying mammograms, US and MRI for breast cancer. Imaging Biomarker Standardization Initiative (IBSI) proposes since 2016 attempts to standardize radiomics features in order to be implemented in daily clinical practice. For a number of 169 radiomic features considered reproducible, it is estimated the verification and calibration of the different software [5,6,7].

## 2. Radiomics—Current Concepts and Future Perspectives

The association of the radiomic features with molecular, genetic and clinical data and their correlation with information regarding the prognosis and the evolution of the oncological disease opens the ways of creating predictive and prognostic models with high accuracy in order to improve the diagnostic and treatment performances in the era of precision medicine. Although the concept of computer-aided diagnosis (CAD) was first reported in 1963, intensive use of CAD has been applied in medical imaging since the last decade of the 20th century. Limited at that time by the computer processing power and also by the reduced data storage capacity, the extraction of data from medical images developed rapidly by increasing the graphic processing and storage capacity of a huge quantity of images. Most radiomics and radiogenomics studies include a limited number of patients, but clinical trials enrolling large cohorts of patients are needed to validate these models in the near future. In this context, it is estimated that both The Cancer Genome Atlas and The Cancer Imaging Archive will play an essential role in radiomics and radiogenomics research in order to validate models with clinical applicability for a precision approach in diagnostic and treatment of cancer [8,9,10,11,12,13,14].

A systematic review analyzing the current radiogenomics data revealed the highest number of studies having as study subject high grade gliomas (35 articles) followed by breast cancer, as a topic of interest in 10 published studies. It is possible that brain tumors are the preferred subjects of radiomic or radiogenomic studies due to the possibility of the tumors’ segmentation which offers a higher capacity of algorithm standardization. Functional imaging methods with diffusion magnetic resonance imaging (DWI) and positron emission tomography (PET-CT) have demonstrated the best ability to correlate radiomics features with the tumor phenotype [15,16,17,18].

The first of the four stages of the radiomic workflow is data acquisition. For this purpose, it is essential to identify a lot of homogeneous features in the study group. The best method for the region of interest (ROI) delineation is also a controversial topic. Too low-resolution ROIs will be affected by image noise, but the delimitation methods vary in the literature, being also a source of uncertainty. The recommended segmented volume varies depending on the imaging method, a value of 5 cmc being recommended as a cutoff value in the case of CT images and 45 cmc in the case of PET-CT imaging. The homogeneity of the acquired images can be influenced by the pixel size, slide spacing, kernel reconstruction algorithm and also by the movement management capability. There is also uncertainty regarding the acquisition times if the contrast agent is used [8,9,10,18,19,20,21,22].

ROI segmentation is the second stage of radiomics that leads to the generation of reproducible images parts proposed for radiomic analysis. Inspired by the manual radiotherapy planning, delineation or segmentation is subject to the variability and physician subjectivity regarding the limits of the tumor tissue and the correct identification of the tumor microenvironment, drawing the barrier between healthy tissue and the tumor being a challenge even in the age of high-resolution medical imaging. Recently, semi-automatic or automatic ROI delimitation algorithms have been developed the potential to reduce errors through inter-observer variability, thus increasing the uniformity of plain images or volumetric delineation. If a single slide is chosen as an option compared to segmenting the entire tumor volume, manual segmentation is more precise but some information that characterizes the entire tumor is thus lost. The delimitation of a single slide and the selection of a tangent circle at the inner edge of the tumor delimited on that slide is also agreed upon in some radiomic studies. The use of low-pass and band-pass filters has demonstrated the possibility to reduce the variability of the evaluated medical images by reducing the number of gray levels. All these improvements in image acquisition and processing have the final goal of obtaining generalizable and highly accurate radiomic models. Stage 3 of the radiomic analysis can be considered as the extraction, processing and reduction of features, so that a finite number of radiomic features are selected, correlated with the purpose of each proposed study [18,23,24].

In the paper “Radiomics: Image has More than Pictures”, the author mentions that the development of information technology has reached the level of progress in which from medical images can be extracted hundreds of variables related to size, shape, texture, in order to be included in the databases for radiomic models creations (the fourth stage of radiomics) [24].

Agnostic features, based on mathematical formulas, are first order intensity hystograms, including features related to voxels: mean, median, energy, entropy, kurtosis, skewness. Other features are related to the shape, volume, surface, characteristics of the sphericity. Among the features related to texture, we mention “Haralick features” which refers to the matrix of gray levels. The variation of a radiomic feature is also evaluated as a potential biomarker of response to therapy, Boldrini et al. for example, considering that this variation called “delta radiomics” may predict response to neoadjuvant therapy in colorectal cancer [25].

Currently, there are approximately 170 features that are analyzed in the creation of radiomic models and several open-source/free software has been created, being capable of extracting features from two-dimensional or reconstructed 3D images. IBEX (Image Biomarker Explorer) and MaZda are examples of computer applications that are able to extract and select significant features from different types of medical images [26,27].

A simplified workflow of radiomics can be divided into two big main steps, the training stage and the applications. A first step that represents the training phase of the workflow aims to associate radiomic characteristics with already known medical data. A higher number of features are initially extracted from diagnostic images via an instrument called radiomics engine. These extracted features are further analyzed in the second platform, called prediction engine, using the machine learning algorithms. The output of the prediction engine transforms the high dimensional feature space of radiomics to a much simpler classification space where the clinical hypothesis of the dataset may be easily assessed. The second step shows how this algorithm can be applied for the prediction of the result on a previously unevaluated medical case. Regarding the methods of analysis based on machine learning, unsupervised machine learning classifies the radiomic features without taking into account information obtained from previous possible correlations of those characteristics with clinical parameters. In this case, the algorithm selects the features based on two data sets (a training and a test data set). In the case of supervised machine learning, the choice of characteristics with statistical semnification will be based on results also validated in the literature. Limiting the number of features in this case can increase the predictive power of the radiomic model, also reducing the need for a large number of subjects in the training set. If an unsupervised machine learning algorithm is chosen, the biggest source of uncertainty is caused by the inhomogeneity of the data obtained from previous research studies due to the lack of standardization. For supervised machine learning, an unsolved problem remains the size and uniformity of the lots for the radiomic models training [23,28,29,30].

Recently the tendency is to use deep learning algorithms that differ completely from the supervised and unsupervised machine learning radiomics analysis. These algorithms direct medical imaging to a multi-layer network similar to a neural network with an ability to reduce the number of features. Subsequently, these features are introduced into a supervised algorithm that produces the predictive model with clinical application [29,30].

## 3. Triple Negative Breast Cancer (TNBC)—A Challenge for the Clinician

TNBC represents about 15–20% of all breast cancers, but the large number of metastatic deaths in this category makes it necessary to understand its subtypes in order to identify effective therapeutic strategies. Being a subtype of cancer that does not benefit from endocrine or anti-HER2 therapy, anthracyclines and taxanes neoadjuvant chemotherapy (NAC) remains the basic treatment. The use of platinum-based chemotherapy is based on the mechanism of action related to DNA damage and is associated with improved pathological complete response (pCR). In the context of medical imagining development, the possibilities of NAC response evaluation have been improved. Thus, the use of NAC as a surrogate biomarker of chemo-sensitivity becomes possible. TNBC pCR after NAC is considered to be associated with superior survival and residual disease after NAC is associated with an unfavorable prognosis. Analyzing gene expression revealed that intrinsic basal-like subtype is the most common triple negative cancer, this subtype being associated with approximately 70% of TNBC cases. Analyzing the genetic profile on 21 data sets consisting of 3247 cases Lehman et al. considers TNBC as divided TNBC into 7 subtypes: 2 basal-like subtypes (Bl1 and BL2), 1 mesenchymal type (M), one immunomodulatory type (IM), 1 mesenchymal stem type (MSI), 1 luminal receptor subtype (LAR) and one type characterized by androgen receptor (AR) signaling. These new subtypes were also confirmed and mentioned in another analysis by Mesuda et al. Although there are predictive differences in the response to NAC and prognostic variations between these groups, this classification cannot be considered as the only factor that modulates the aggressiveness of TNBC. The present BRCA1 mutation is associated with DNA sensitivity to platinum-based chemotherapy or to poly (ADP-ribose) polymerase (PARP) inhibitors due to homologous recombination (HRD) deficiency. Another positive prognostic factor is the presence of Tumor-Infiltrating Lymphocytes (TILs) > 50% for the immunomodulatory type, lymphocyte infiltrate being considered the TNBC pattern with the best prognosis in the absence of any treatments. Furthermore, the presence of TILs in the tumor after NAC is associated with a more favorable evolution. The LAR subtype is enriched in AR and all cases of LARs exhibit a mutation in the kinase domain of PIK3CA, exhibiting a high sensitivity to PIK3CA inhibitors [31,32,33,34].

Analyzing TNBC Lehmann subtypes, Santonja et al. demonstrates that the LAR subtype is the least proliferative and manifests the highest chemoresistance but this fact is not a negative prognostic factor if it is associated with AR-positives. The results regarding the prognosis of the LAR subtype, however, remain controversial, being associated with both the most favorable and the most unfavorable prognosis in different clinical trials. As evidenced by the Phase II trials, the TNBC subset with positive AR will benefit from antiandrogens treatment [35,36].

The BL1 subtype is characterized by sensitivity to chemotherapy, including platinum agents. Even if the results of the phase II studies are controversial, the homologous recombination deficiency (HRD) has been associated with a higher response to platinum agents. The BL2 subtype is reached in angiogenic factors, so it may respond to treatment with vascular endothelial growth factor receptor (VEGFRs) inhibitors, Orantinib, an anti-angiogenic agent that targets other receptors in this subtype, demonstrating a favorable response in combination with docetaxel in cancer cases of metastatic breast showing anthracycline resistance. Subtype BL2 has the lowest pCR after neoadjuvant chemotherapy [35,37,38].

The triple negative breast cancer represents one of the most interesting fields of interest in research, considering the lack of therapeutic resources that will lead to a significant improvement of the prognosis of this subtype of the disease [39,40].

TNBC is a heterogeneous disease with numerous subtypes involved in prognosis and response to different types of therapies. Recent research concerns have taken into consideration the evaluation of the subtypes that benefit from platinum-based chemotherapy and for which standard anthracycline and taxane-based chemotherapy offers a maximal response. Both PARP inhibitors in cases with mutant BRCA and androgen receptor inhibitors, treatment with sacituzumab govitecan (an antibody-drug conjugate) and PI3K/AKT and PD-L1 inhibitors are among the topics of interest. Sacituzumab Govitecan, a metabolite of irinotecan, has shown promising results in the ASCEND Phase III trial, with response rates of 33% in TNBC patients treated intensively. Even if PARP inhibitors have already shown efficacy in the case that germline BRCA mutation is detected, the OLYMPIA trial will elucidate the role of PARP inhibitors in BRCA mutant tumors and stratify patients who will benefit from this treatment. Immunotherapy is becoming an increasingly important component of therapeutic associations with PARP inhibitors. Based on the idea that PARP inhibitors in combination with checkpoint inhibitors modulate response to tumor microenvironment therapy, the therapeutic associations between immunotherapy and PARP inhibitors are intensively studied. The KEYNOTE-022 trial evaluates the association of the efficacy of the addition of pembrolizumab to standard non-addictive chemotherapy based on platinum and taxanes. Combination with veliparib, a PARP inhibitor compared to chemotherapy, did not benefit the pCR rate, but carboplatin AUC6 demonstrated superiority in pCR in the BrigTness trial. However, these strategies remain options in the recurrent or metastatic settings, and chemotherapy still remains the basis of systemic treatment in TNBC [41,42,43,44].

Several promising new treatment options are undergoing active evaluation in many clinical trials. Among these strategies such as PARP, MEK, AKT pathways inhibition and the association of checkpoint inhibitors with chemotherapy, the most promising research method include radiomic analysis. In this context, non-invasive identification of treatment response predictors for TNBC identified from medical images becomes a priority. Identifying valid and accessible biomarkers to predict the benefit of each therapeutic strategy is one of the directions with the greatest impact in the treatment of TNBC to which radiomics can make a valuable contribution [45,46].

Experimental preclinical models can accelerate new molecular target identification, thus improving therapeutic efficacy. Patient-derived xenografts (PDX), due to their genomic and transcriptomic fidelity to the tumors from which they are derived, are able to improve preclinical testing of target drug combinations in translational models. Despite previous development of breast PDX and TNBC models, those derived from patients with demonstrated triple negative breast tumors are lacking.

The use of hybrid imaging in radiomic analysis was applied on a PDX model by TNBC in order to implement this method in clinical practice. With the help of preclinical models developed on mice, concepts based on radiomics that can be implemented in daily clinical practice will be proposed. The limited utility of tumor cell lines established in refining preclinical models in the context of new knowledge related to tumor heterogeneity has made it necessary to use transplantable tumor xenografts, PDX models being superior for this purpose. A study that analyzed gene expression of 93 TNBC in order to correlate it with the five subtypes of TNBC aimed to evaluate the reproducibility of radiomic models in PET-CT imaging in preclinical settings. The method included evaluation of tumors with volumes varying between 100 and 450 mm^3^, by the PET-CT image parameters acquisition using 18F-fluorodeoxyglucose (18F-FDG) at 45–60 min post-injection. The analysis of the results also aims the standardization of the data acquisition and features extraction in order to correctly identify the five subtypes of TNBC using solid and reproducible radiomic models [47,48].

## 4. Radiomics in Breast Cancer Research

Breast cancer is the most commonly diagnosed cancer among women and the second leading cause of cancer death. In an ERA when medical imaging is increasingly used for diagnostic and evaluation of response to oncological therapy, identifying prognostic and predictive biomarkers of post-therapeutic evolution is becoming increasingly important [39].

The development of radiomics, which can extract quantitative characteristics from the volume of interest (VOI) delineated from medical images including shape/size, intensity and texture, has also benefited from translational research in breast cancer. In the case of breast cancer, radiomic analysis demonstrated the ability to differentiate malignant tumors from benign tumors, to identify molecular subtypes, to predict certain radioresistance and chemoresistance factors, to predict malignancy and to anticipate early the response to neoadjuvant chemotherapy (NAC) [49].

Radiomic analysis was initially validated in lung cancers, but the need for an accurate diagnosis with the reduction of false positive US, mammography, digital breast tomosynthesis (DBT) and MRI examinations are methods desirable in breast cancer radiomics research. An increased predictive performance was demonstrated using tumor entropy as a feature extracted from MRI imaging and DBT, a new technology with promising results especially in the case of women with dense breasts. Implications of radiomics for breast cancer screening are major given the anxiety effect of a potential cancer diagnosis and a useless biopsy [40].

Avanzo et al. performed a systematic review focused on the use of artificial intelligence (AI) in medical imaging, based on scientific articles published in Italy. The authors also assessed the areas of interest in which AI was used. In a second phase of the analysis, the AI4MP working group dedicated to artificial intelligence from the Italian Association of Physics in Medicine (AIFM) also performed an analysis of the results from the literature. Excluding reviews and articles published outside the 2015–2020 period, the authors identified 122 publications in Pubmed and 46 studies in the working group. Cancer was the subject of 25% of AI studies and MRI was the most commonly used diagnostic method for radiomics. The authors note a significant increase in interest regarding AI in medical imaging, especially after 2018 [50].

The Italian MAGIC-5 trial aims to implement a database that facilitates the diagnosis of a CAD algorithm, based on the digitization of mammography images. Within the project, 3369 images obtained from 967 patients were digitized. The classification of images was based on the type of lesion, morphology and pathology of the breast, also developing a graphical interface to facilitate image analysis and their use as a basis for radiomic analysis or classification using neural networks, but also for epidemiological studies. Given the large amount of data collected in screening projects from different geographical areas, expanding and interconnecting the network will be a challenge for the future [51].

In an attempt to eliminate the operator-dependent factor that affects the diagnostic capacity of Contrast-Enhanced Spectral Mammography (CESM), Fanizzi et al. propose a CAD, based on the features extraction from low-energy and recombined images. The study used 48 ROIs obtained from 53 patients, evaluated by previously trained Random Forest type algorithms. The authors select only significant radiomic features for the building of the diagnostic model. The radiomic algorithm demonstrates an improved diagnostic capacity by 8% compared to the human subjective examiner, the specificity and selectivity being 87.5% and 91.7%, respectively. Massafra et al. analyzed 58 ROIs from 53 patients extracting 464 radiomic features, including both characteristics obtained from original ROIs and Haar decomposition image gradient. With a specificity of 88.37% and a selectivity of 100%, the authors demonstrate the ability of the radiomic model to discriminate between benign and malignant tumors. In this case, the Principal Component Analysis (PCA) was used to reduce the dimensions in the classification scheme [52,53,54].

Radiomics was also proposed as a method for early evaluation of the oncological neoadjuvant therapy results, with background parenchymal enhancement (BPE) parameter from breast MRI images being evaluated for the predictive potential. Thus, the authors want to evaluate the ability of BPE to become an imaging biomarker with a predictive role of the response to therapy in breast cancer. The study benefited from the experience of three expert imagist physicians who evaluated MRI images obtained from 80 patients before and after the completion of chemotherapy, but also at 3 months after surgery. BPE reduction was significantly associated with the administration of the first anthracycline/taxane-based chemotherapy cycle, being more significant than in the case of anti-HER2 therapy. Another factor associated with BPE reduction after initiating neoadjuvant therapy was tumor size. Analyzing the BPE obtained from breast MRI images, La Forgia and collaborators highlight the potential of the method to predict early the response to therapy but also mention the potential of including radiomic analysis in the algorithm for early prediction of response to neoadjuvant therapy in breast cancer [55].

## 5. Radiomics and Breast Cancer Imaging Methods: A Brief Comparative Assessment

Medical imaging is essential both in diagnosis and staging and in treatment planning and monitoring during the course of the disease. At this time all imaging methods involved in the management of breast cancer are associated with radiomic analysis. In an analysis that synthesizes data regarding the use of radiomics in breast cancer focused on MRI, mammography and DBT, Lee and collaborators identify a majority (16 studies) from a total of the 25 radiomic studies considered representative for breast cancer. Four studies used features extracted from the US, four were associated with mammography and one with DBT. The number of features extracted varied greatly (from 45 to 13,950). Entropy, textural and gray level matrix features are the most commonly used radiomic features. The radiomic score (rad-score) and the radiomic nomograms are mentioned in two and three studies, respectively. Of note are the studies of Braman et al. who approach the identification of radiometric features extracted from peritumoral DCE-MRI to predict the response to NAC and anti-HER2 + target therapy for the HER2 + subtype. Four of the twenty-five studies aim to predict the response to therapy and five studies aim to predict therapeutic failure by lymph node metastasis (three studies) and local recurrence (two studies). Radiomic and Rad-score nomograms were more commonly used in predictive models than in diagnostic ones [56,57,58].

The use of CT imaging in radiomic analysis related to breast cancer is associated with a relatively small number of studies, probably justified by the limited value of CT in diagnosis and the superior sensitivity of MRI and PET/CT in the detection of lymph node metastases. However, Yang’s study demonstrates 89.1% and 88.5% accuracy in assessing lymph node metastases in the test, respectively, of the validation cohort in a radiomic study using CT imaging. Piñeiro-Fiel evaluates 20 publications that propose the use of PET-associated radiomics in breast cancer, most of them (75%) extracting images from PET images alone, and the rest of the studies being dedicated to hybrid imaging (PET-MRI (20%) and PET-CT (5%)). The study by Antunovic et al. proposes PET-CT radiomics for the prediction of pCR after NAC for patients with locally advanced breast cancer. The analysis also identifies HER2 + and TNBC subtypes as being associated with an unfavorable response. Roy et al. proposes a radiomic analysis on patient-derived tumor xenografts (PDX) to optimize the radiomic signature so as to obtain a robust FDG-PET signature in order to predict response to therapy. In contrast to MRI, DBT and mammography studies, in this case of PET radiomics, only 20% of the studies aim at diagnosis and tumor staging. Most cases in which PET radiomics is proposed for the management of breast cancer (45%) are associated with the prognostic correlation of extracted features or feature sets [59,60,61].

## 6. Radiomics and TNBC

### 6.1. TNBC Molecular Differential Diagnosis

An association method study investigated the role of optoacoustic imaging combined with gray-scale ultrasound (OA/US) for molecular breast cancer subtypes detection. Using the Kruskal–Wallis test, US and OA/US the molecular subtypes of breast cancer were identified. With the support of two methods—OA analysis combined with the US grayscale, the difference between luminal subtypes and TNBC and HER2 was evaluated. The same favorable results in differentiating between the TNBC and HER2 positive breast cancer were obtained in a retrospective analysis by Dogan and collaborated on a lot of 532 cases of breast cancers in 519 patients with molecular markers available [62,63].

Radiomic signatures extracted from contrast-enhanced magnetic resonance imaging (EC-MRI) also demonstrated the possibility of evaluating the status of breast cancer receptors in a study of 143 breast cancer patients for assessing breast cancer receptor status and subtypes. Tumors were manually segmented and subsequently the radiomic analysis included features such as first-order histogram co-occurrence matrix, run-length matrix, absolute gradient and tumor geometry. Significant radiomic features were selected and the patients were divided into a training set and a validation set. The radiomic analysis proved an accuracy of 79.4% for the differentiation of the luminal A and luminal B subtypes. For the luminal B differentiation from TNBC the accuracy of the method was 77.1%. The authors conclude that breast CE-MRI imaging may be the basis of radiomic analysis in order to detect the molecular subtypes of this type of cancer [64].

Another study aims to determine the value of the surrounding tumors parenchyma enchancement included in the analysis, simultaneously with tumor analysis of DCE-NMR images. Identifying images obtained from 84 women diagnosed with 88 invasive breast carcinomas, the authors proposed the evaluation of the images by both an expert imagist physician and by radiomics analysis. In total, 85 features extracted from the tumor and from the surrounding parenchyma were analyzed. A reduced number of radiomic features were selected as significant and a predictive model based on support vector machines was created. The radiomic features extracted from the tumor region images as the only method of evaluation proved inferiority compared to the case where the parenchyma enchancement characteristics were also analyzed. Among the most useful features for predicting TNBC were the textures and heterogeneity of background parenchymal enhancement [65].

To test the power of a radiomic model in making a differential diagnosis between TNBC and non-TNBC breast cancer using preoperative CT, a retrospective study was proposed that included 200 non-TNBC patients and 100 TNBC patients. Of these, 180 cases were used in the discovery cohort and 120 in the validation group. Five radiomic features were identified as predictive to discriminate the TNBC subtype from non-TNBC. The mean predictive value for TNBC was identified as 0.881 and 0.851 in the discovery case group and in the validation group, respectively. The study demonstrates the ability of a radiomics signature based on preoperative CT imaging to identify TNBC from other molecular subtypes of breast cancer. Using a radiomic signature based on 15 quantitative features extracted from DCE-MRI images of breast cancer, Ma and collaborators were able to make a differential diagnosis of TNBC/non-TNBC. In this study, ROIs were automatically segmented by a deep learning algorithm and subsequently validated by two radiologists [66,67].

Based on radiomic features extracted from images from 120 breast cancer patients and manually delineated tumors (90 non-TNBC and 30 TNBC), Zhang et al. were able to identify TNBC and differentiate it from other types of breast cancer by identifying four radiomic features (roundness, concavity, gray average and skewness) that are different between TNBC and non-TNBC breast cancers. Triple negative tumors are more round and regular (less concave), with increased gray density and low skewness [68].

### 6.2. Differentiation between TNBC and Fibroadenoma

Differential diagnosis of TNBC with fibroadenoma is sometimes difficult, TNBC often presents in breast ultrasound a benign morphology. A study that aims to use the US for differential diagnosis between TNBC and fibroadenoma by radiomic analysis included 715 fibroadenomas, 186 TNBC, all cases pathologically confirmed. In total, 730 features were extracted from acquired images (14 intensity-based features, 132 textures features and 584 wavelet-based features). Radiomic analysis highlights favorable results, even for lesions in the three or four BI-RADS category, probably considered benign or suspected to be malignant by expert imaging physicians. In the future, after overcoming the problems related to standardization, radiomic analysis is expected to reduce the number of biopsies for false-positive imaging diagnoses [69].

Another study that aims to evaluate radiomics analysis in the differential diagnosis uses images from 169 tumors (84 benign fibroadenomas and 85 TNBC tumors). After tumor segmentation using the level-set method, radiomic texture features are analyzed. Both conventional morphology and multiresolution gray-scale invariant texture feature were evaluated in the study. Computer-assisted diagnostics (CAD) uses best-fitting ellipse, gray-level co-occurrence matrices and the ranklet transform methods. The authors concluded that the use of radiometric texture features extracted by ranklet transformatiom can be successfully used in differential diagnosis between fibroadenomas and TNBC [70].

### 6.3. Prognosis and Prediction of Response to Neoadjuvant Chemotherapy

Another research direction in TNBC radiomics is the evaluation of the method’s ability to analyze intratumoral and peritumoral regions using DCE-NMR, in order to predict the complete pathological response (pCR) to NAC. For this purpose, 117 patients who received NAC, were included in the study that analyzed the intratumoral and peritumoral regions of DCE-NMR scans in T1 sequence. After VOI segmentation, 99 radiomic textural features were subsequently extracted. The training set included 78 images aiming at the training of multiple machine learning classifiers, in order to predict the probability of pCR for other newly diagnosed patients. The evaluation lot included 39 patients. In a second stage of the study data related to HR status and HER status were included.

The results highlighted the concept that a set of combined intratumoral and peritumoral radiomic features lead to superior results, the HR+, HER2, non-pCR subtypes being characterized by higher peritumoral heterogeneity during initial contrast enhancement. TNBC and HER2 + tumors are associated with an enhancement pattern within the peritumoral regions for the therapy non-responder cases. The authors concluded that independently of the classifier choice the radiomics analysis of the DCE-NMR images demonstrates the possibility of predicting the response to NAC before the treatment whether or not we know the status of the receptors [58].

Until the study of Koh et al., dynamic contrast-enhanced radiographic (DCE) MRI analysis of the entire tumor (three-dimensional) is not known to have been used to evaluate the prognosis of TNBC. This study aims not only to identify the prognostic power of the radiomic model proposed, but also to validate it for various types of MRI scanners. Data were extracted from 231 cases of TNBC ranging in age from 23 to 88 years. Out of a total of 3995 radiomic features through 10 fold cross validation for LASSO, 32 features were finally selected and a radiomic score (Rad-score) was generated. It should be noted that seven features were selected from the gray level co-occurrence matrix (GLCM), 24 from gray level run length matrix (GLRLM) and 1 from histogram analysis. The size of the pathological invasion, lymphovascular invasion, number of metastatic axillary nodules and type of surgery are features identified as predictive in the clinical model. The Rad-score and the clinical model were more accurate in predicting the clinical model than the clinical model alone, but external validation on other MRI scanners did not confirm the superiority of the radiomic over the clinical model [71].

A multicenter study based on radiomic features extracted from the US that included 486 TNBC cases proposed the validation of a nomogram that included a radiomic Rad-score and clinicopathological criteria to predict DFS after surgery. Elevated values of the proposed Rad-score from greyscale ultrasound features were correlated with worse outcomes. The nomogram had superior predictive power to the clinicopathological marrow and the number-based prediction of metastatic lymph nodes. The authors conclude that the addition of clinicopathological data to the radiomic model increases the power of prediction compared to any of the independently evaluated models [72].

Early recurrence after NAC in TNBC was successfully predicted by the radiomic model based on pre- and post-NAC MRI imaging. Including 147 patients divided into the training set (104 cases) and the test set (43 cases) and a deep learning algorithm for automatic tumor segmentation from contrast enhanced MRI images, the study proposed 3 radiomic models (pre-chemotherapy, post-chemotherapy and combined) and a model with clinical parameters. Recurrence 3 years after NAC was accurately accurate when using both pre- and post-chemotherapy features [73].

The correlation between radiomic features and the level of Tumor-Infiltrating Lymfcite (TILs) identified in patients diagnosed with TNBC proposed by Yu and colleagues aims to identify the early identification (from mammography images) of predictors of response to chemotherapy and immunotherapy in TNBC. 204 features including morphological, textural, and greyscale features were extracted from tumors delineated on mammogram images and statistically correlated with tumor TILs. A cutoff value of 50% was used to evaluate the “low” and “high” values of the TILs. The study identified 6 significant radiomic features (unformity, variance, and four gray matter matrix-related features) between “low TILs” and “high” TILs. The prognostic and predictive value is given by the concept that tumors with “high” TILs are associated with a higher rate of complete pathological response and higher survival rates [74].

The main radiomics studies involving both different non-medical imaging methods and different clinical endpoints have been summarized and presented in a table (Table 1).

## 7. Conclusions

Radiomics, a relatively new but promising translational research field, adds value to precision oncology, with TNBC already benefiting from the results. Differentiation between fibroadenoma and TNBC, identification of molecular subtypes from US, MRI and other medical imaging and early prediction of response to NAC is already validated applications of radiomics. Future research in radiomics with major clinical applications would target sensitivity to platinum-based chemotherapy, target therapies and immunotherapy response prediction, in order to improve the prognosis of this aggressive breast cancer subtype. Thus, we estimate that radiomics will become a key player in the precision oncology of TNBC. The correlation between radiomics markers, response to different therapeutic protocols, prognosis and molecular subtypes of TNBC, radiomics will offer the perspective of treatment personalization with minimal costs. Already having clinical results regarding the benefit of combination platinum-based chemotherapy-PARP inhibitors—immune checkpoint inhibitors, radiomics models can provide predictive biomarkers of therapeutic response in order to stratify the oncological therapy and to offer in each case of TNBC the best therapeutic option. Deep learning algorithms tend to replace the classic supervised and unsupervised radiomics, but radiomics is easier to implement and we estimate that the method will still play an important role in translational cancer research. In the ERA of “big data” we expect the rapid initiation of studies that will translate radiomics into the current clinical oncology practice of TNBC. If radiomics combining standard screening and diagnostic methods seems to be more focused on differential diagnosis and staging, MRI, PET and hybrid PET-CT and PET-MRI in association with radiomics score and nomograms tend to be key players in precision diagnosis, contributing to the stratification of patients and the identification of potential candidates for each therapeutic agent, being key elements in future precision TNBC therapy. Radiomic analysis, especially by MRI and mammography, is estimated to improve the specificity and maintain a high rate of diagnostic sensitivity, thus reducing the necessary biopsies, but will also open new horizons in the precise selection of patient groups, candidates for the omission of surgery after complete response to NAC. 

## Figures and Tables

**Table 1 jcm-11-00616-t001:** Radiomics for TNBC: the table includes the imaging method chosen for radiomic analysis, the type, the number of features used in creating the model including the radiomic score and nomograms, study objectives, authors and year of publication [58,62,63,64,65,66,67,68,69,70,71,72,73,74].

Imaging Method	Radiomic Features/Features Number/Radiomic Signature	Study Objective	
US	optoacoustic imaging (OA) combined with gray-scale US	identify the differences between molecular subtype	Menezes et al. (2019) [62].
MRI	first-order histogram (HIS), co-occurrence matrix (COM), run-length matrix (RLM), absolute gradient (GRA), autoregressive model (ARM), discrete Haar wavelet transform (WAV), and lesion geometry (GEO)	asessment of breast cancer receptor status and molecular subtypes.	Leithner et al. (2019) [64].
MRI	85 radiomic features (morphologic, densitometric, texture)	distinguish triple-negative cancers from other subtypes	Wang et al. (2015) [65].
CT	radiomic signature based on preoperative CT	guidance in choosing the treatment	Feng et al. (2020) [66].
MRI	15 features	to differentiate triple-negative breast cancer (TNBC) and nontriple-negative breast cancer (non-TNBC).	Ma et al. (2021) [67].
X-ray mammography	roundness, concavity, gray average and skewness	distinguish between TNBC and non-TNBC	Zhang et al. (2019) [68].
US	730 features (14 intensity-based features, 132 textural features and 584 wavelet-based features)	differential diagnosis between triple-negative breast cancer and fibroadenoma	Lee et al. (2018) [69].
US	morphology, conventional texture, and multiresolution gray-scale invariant texture feature	distinguishing between TNBC and benign fibroadenomas	Moon et al. (2015) [70].
MRI	both peritumoral and intratumoral features	prediction of pathological complete response (pCR) to neoadjuvant chemotherapy (NAC).	Braman et al. (2015) [58].
MRI	Rad-score	prediction of systemic recurrence	Koh et al. (2020) [71].
US	Rad-score and radiomic nomogram	prediction of disease-free survival	Yu et al. (2021) [72].
MRI	Three radiomic models based on pre- and post-NAC magnetic resonance images	prediction of systemic recurrence after NAC	Ma et al. (2022) [73].
Mammography	radiomics nomogram that incorporates Rad-score	prediction of invasive disease-free survival	Jiang et al. (2020) [74].

## Data Availability

Not applicable.

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
