# Peer review of "Radiomics in Triple Negative Breast Cancer: New Horizons in an Aggressive Subtype of the Disease"

_jcm, 2022, doi:10.3390/jcm11030616_

Round 1

Reviewer 1 Report

The present study is a review of the role of radiomics in triple negative breast cancer. The manuscript is interesting and well structured. I suggest summarizing some parts as it risks being repetitive on some points. I suggest adding tables to summarize the most relevant studies reported in the specific topic.

Author Response

Dear Reviewer,

Thank you for evaluating and correcting the manuscript. At your suggestion, we have included a single table showing the main studies of TNBC, considering that the inclusion of other tables would dilute the focus of TNBC on radiomics in breast cancer or on some imaging aspects. We also removed some repetitive ideas about sacituzumab govitecan. I also added to the suggestion of reviewer 2, two ideas for conclusions about future directions for omitting biopsy by increasing diagnostic accuracy and omitting surgery for complete responders after NAC. Hoping you will appreciate these changes, we are waiting for new suggestions.

Best regards,

Camil Mirestean

Reviewer 2 Report

at page 3 you describe the first of 4 process in radiomics but it would be useful to read the others numbered.

at page 6 you repeat the same item about  s. govitecan

would be interesting in the conclusions  to better define future perspectives as ongoing studies in which radiomics could demonstrate how to improve clinical processes ( if possible) as avoiding biopsy or surgery after neoadjuvant treatment 

Author Response

Dear Reviewer,
Thank you for evaluating and correcting the manuscript. At your suggestion, we mentioned in the right place the other 3 stages of radiomic analysis and we also removed some repetitive ideas about sacituzumab govitecan. I also added
two ideas to conclusions about future directions for omitting biopsy by increasing diagnostic accuracy and omitting surgery for complete responders after NAC. At the suggestion of reviewer 1 asm added a table with the main radiomics studies in TNBC. Hoping you will appreciate these changes, we are waiting for new suggestions.

Best regards,

Camil Mirestean